# Anti-Inflammatory Effect of an *O*-2-Substituted (1-3)-β-D-Glucan Produced by *Pediococcus parvulus* 2.6 in a Caco-2 PMA-THP-1 Co-Culture Model

**DOI:** 10.3390/ijms23031527

**Published:** 2022-01-28

**Authors:** Sara Notararigo, Encarnación Varela, Anna Otal, María Antolín, Francisco Guarner, Paloma López

**Affiliations:** 1Molecular Biology of Gram-Positive Bacteria, Margarita Salas Center for Biological Research (CIB-Margarita Salas-CSIC), Department of Microbial and Plant Biotechnology, Ramiro de Maeztu 9, 28040 Madrid, Spain; sarita.not@hotmail.com; 2Digestive System Research Unit, Institut de RecercaValld’Hebron (VHIR), University Hospital Valld’Hebron, Universitat Autònoma de Barcelona, 08193 Barcelona, Spain; encarna.varela@vhir.org (E.V.); aotal88@gmail.com (A.O.); maria.antolin@vhir.org (M.A.); francisco.guarner@vhir.org (F.G.); 3Foundation Health Research Institute of Santiago de Compostela (FIDIS), 15706 Santiago de Compostela, Spain; 4CIBERehd, Instituto Carlos III, 28029 Madrid, Spain

**Keywords:** (1-3)-β-D-glucan, bacterial exopolysaccharides, immunomodulators, *Pediococcus parvulus*, post-biotic, anti-inflammatory

## Abstract

Bacterial β-glucans are exopolysaccharides (EPSs), which can protect bacteria or cooperate in biofilm formation or in bacterial cell adhesion. *Pediococcus parvulus* 2.6 is a lactic acid bacterium that produces an *O*-2-substituted (1-3)-β-D-glucan. The structural similarity of this EPS to active compounds such as laminarin, together with its ability to modulate the immune system and to adhere in vitro to human enterocytes, led us to investigate, in comparison with laminarin, its potential as an immunomodulator of in vitro co-cultured Caco-2 and PMA-THP-1 cells. *O*-2-substituted (1-3)-β-D-glucan synthesized by the GTF glycosyl transferase of *Pediococcus parvulus* 2.6 or that by *Lactococcus lactis* NZ9000[pGTF] were purified and used in this study. The XTT tests revealed that all β-glucans were non-toxic for both cell lines and activated PMA-THP-1 cells’ metabolisms. The *O*-2-substituted (1-3)-β-D-glucan modulated production and expression of IL-8 and the IL-10 in Caco-2 and PMA-THP-1 cells. Laminarin also modulated cytokine production by diminishing TNF-α in Caco-2 cells and IL-8 in PMA-THP-1. All these features could be considered with the aim to produce function foods, supplemented with laminarin or with another novel β-glucan-producing strain, in order to ameliorate an individual’s immune system response toward pathogens or to control mild side effects in remission patients affected by inflammatory bowel diseases.

## 1. Introduction

The human intestinal tract consists of a unique tubular tissue structure supported by various cell types located in different layers. The internal lumen is completely covered by a complex epithelium, which has in its apical region the microvilli, which increases the surface area for absorption of nutrients and other compounds. Each layer of the intestine has a significant diversity. While the epithelium consists mainly of enterocytes, which are responsible for absorbing and transporting nutrients, there are three other cell types that carry out secretory functions, including goblet cells, Paneth cells and enteroendocrine cells. In addition, in the crypts of Lieberkühn, there are intestinal stem cells responsible for the high regenerative capacity of the epithelium, which is renewed approximately every 7 days [1,2].

Since the gut is so complex, several in vitro models have been developed to study the effect of microorganisms, molecules and compounds present in the lumen [3,4]. The most widely used is the so-called in vitro intestinal barrier model, formed by a monolayer of primary culture of human intestinal epithelial cells (with mainly absorbent characteristics). This allows the study of permeability and the passage of substances through the barrier. Although human enterocytes retain important anatomical and biochemical characteristics during the in vitro culture, their use also has major drawbacks, due to their culture difficulties and limited viability, as well as their inability to form polarized epithelial monolayers typical of a tissue enterocyte that shows an apical and a basolateral surface. Therefore, human carcinogenic cell lines such as Caco-2 or HT-29, (as well as its mutant variant HT-29 MTX) [5], derived from a colorectal carcinoma [6,7], are routinely used to establish in vitro intestinal models for pharmacological and pharmacokinetic application to evaluate strategies for investigation drug absorption [8] or human goblet cell differentiation and mucin secretion [9,10].

Both types of cell lines have been also used to study the adhesive capacity of lactic acid bacteria (LAB) [11,12,13] and Bifidobacterium [14,15] that produce exopolysaccharides (EPSs). However, this model represents only the feature of the enterocytes, but not the complex interactions that take place between the cells that constitute the gut epithelium and the resident gut immune system cells. In addition, the Caco-2 cell line represents the normal mucosa, without considering the pathophysiologic changes that occur under inflammatory conditions, stress or tissue damage. Therefore, they are not suitable for the study of the effects of drugs or adjuvants during pathological processes. To cover these deficiencies, different in vitro models of intestinal barrier co-cultured with immunocompetent cells that emulate the structure and conditions of the intestine in vivo have been devised [4]. Initially the co-cultured model of inflamed intestinal mucosa was developed using epithelial cell lines along with primary cultures of macrophage and/or dendritic cells to mimic lamina propria compartmentalization [16]. However, the use of immune cell lines is now well established to avoid the drawbacks of such cultures. Thus, co-culture of Caco-2 cells and the macrophage-like cells PMA-THP-1 constitute an ideal model to study signal transduction between epithelial cells and macrophages under the action of various stimuli [17,18,19].

It has been shown that the presence of the bacterial *O*-2-substituted (1-3)-β-D-glucans increases the adhesion capacity of LAB producers to human Caco-2 epithelial cells [11]. It is noteworthy that the beneficial effect of this type of homopolysaccharide (HoPs) is not shared by many of the bacterial EPSs studied (HoPs and heteropolysaccharides (HePs)). For example, the ability of Lb. johnsonii FI9785 to adhere to intestinal epithelial cells decreases when the production of its HePs is reduced [20]. Likewise, it has been demonstrated that, for *Lacticaseibacillus rhamnosus* GG (formerly *Lactobacillus rhamnosus* GG), the presence of its EPS (rich in galactose) on its surface decreases its degree of adhesion [21,22]. Moreover, HoPs such as dextrans have different influences, decreasing, increasing or not affecting binding to Caco-2 cells of strains belonging, among others, to the Lactobacillus, Leuconostoc and Weissella genera [23,24,25].

Previous results revealed that *O*-2-substituted (1-3)-β-D-glucans synthesized by LAB are able to immunomodulate macrophages derived by human monocytes and PMA-THP-1 under lipopolysaccharide (LPS) stimulus by a consistent reduction of pro-inflammatory mediators [26]. Moreover, Notararigo et al., (2021) [27] and Perez-Ramos et al., (2018) [28] demonstrated that *O*-2-substituted (1-3)-β-D-glucan reduced chemokine IL-8 in an ex vivo model of human intestinal biopsies and in vivo zebra fish model.

Additionally, beneficial health influences of the *O*-2-substituted (1-3)-β-D-glucans have been detected including pre-biotic as well as hypocholesterolemic effects. This conclusion is supported by testing of the EPS included in or synthesized in situ in food: (i) in vitro by treatment of probiotic LAB [29,30] and (ii) in vivo using (a) mouse models [31,32] including a model of atherosclerosis [33] and (b) in human trials [34].

Taken together, all these previous results indicate that *O*-2-substituted (1-3)-β-D-glucans may be able to modulate the intestinal immune response and that they and/or the producing bacteria can be useful as components to develop functional food.

Thus, in this work, an in vitro transwell model has been established to investigate the immunomodulatory effect of (1-3)-β-D-glucans, substituted in position *O*-2 (bacterial EPS) or *O*-6 (laminarin), due to their interaction with the intestinal epithelial mucosa, and to study the effects of these treatments on cellular mechanisms.

## 2. Results

### 2.1. Effects of (1-3)-β-D-Glucans on Caco-2 Cells Viability

Prior to establish a transwell inserts model, the potential toxic effect of the (1-3)-β-D-glucans synthesized by the pediococcal GTF glycosyl transferase in *P. parvulus* 2.6 (EPS P) or the recombinant *Lactococcus lactis* strain NZ9000[pGTF] (EPS L) and laminarin on the viability of Caco-2 cells was evaluated. The effect of the laminarin was also tested in this and subsequent experiments, because it is an immuno-stimulant and, like the EPS, is a (1-3)-β-D-glucan, but with branches in the *O*-6 position instead of in the *O*-2 position.

The results revealed that none of the treatments affected the cellular metabolism of Caco-2 cells neither during their differentiation stage (high proliferation rate, Figure 1a) nor when the monolayer was already differentiated and polarized (non-proliferative rate, Figure 1b).

Moreover, it was observed that both bacterial (1-3)-β-D-glucans showed a tendency to increase cellular respiration in both conditions, even though results were not statistically significant (Figure 1a,b). On the contrary, laminarin showed more homogeneous values compared to the EPS (Figure 1a,b). Beside these differences, these results confirmed that the EPS as well as the laminarin were not toxic to Caco-2 cell cultures during any of the differentiation phases tested.

### 2.2. Effects of (1-3)-β-D-Glucans on the Viability of PMA-THP-1 Cells

A potential toxic effect of the EPS and of laminarin on PMA-THP-1 cells was also investigated. The results are depicted in Figure 1c and showed no detrimental effect of any of the three (1-3)-β-D-glucans on PMA-THP-1 cell viability. The percentage of survival was not lower than that of the untreated control (100%) at any of the concentrations of the EPS or the laminarin tested. All the treatment increased cellular respiration in PMA-THP-1 macrophages, as depicted in Figure 1c. EPS L showed a significant difference at the concentration of 10 μg mL^−1^ and 50 μg mL^−1^, EPS P at the concentration of 2, 10, 25 and 50 μg mL^−1^, and laminarin from 2 to 75 μg mL^−1^, with a *p* < 0.05.

### 2.3. Establishment of a Co-Culture Model with Caco-2 and PMA-THP-1 Cells under LPS Stimulation

Transwell inserts were used to establish an in vitro model of the gut mucosa, employing Caco-2 cells differentiated to enterocyte in the upper chamber and PMA-THP-1 macrophages in the lower one.

To test the functionality of the in vitro co-cultured model, LPS was added in the lower compartment, where PMA-THP-1 where cultured. Then cytokine profiling of TNF-α and IL-8 (pro-inflammatory) and IL-10 (anti-inflammatory) was carried out in the Caco-2 and PMA-THP-1 compartments (Figure 2 and Figure 3). Two treatment endpoints were considered: after 6 h, (Figure 2) and after 24 h (Figure 3), in order to assay the effect of LPS on signal transduction between the two cell types that constitute the gut mucosa co-culture model.

After 6 h, LPS addition resulted in a significant increase of TNF-α, IL-8 and IL-10 concentrations in both reservoirs containing either Caco-2 (Figure 2b) or PMA-THP-1 (Figure 2c) cell lines.

However, after 24 h of treatment, the two cell lines responded differently to the stimulus. The Caco-2 secreted only a significative amount of IL-8 (approximately 2000 pg mL^−1^), which was a 4-fold increase vs. CTRL/CTRL. TNF-α reached a concentration of 135 pg mL^−1^, which corresponded to a 3-fold increase vs. CTRL/CTRL, while no significant effect was observed for IL-10 levels compared to the untreated control (Figure 3b). Concerning the cytokine production by PMA-THP-1, upon exposure to LPS, significant increased values for IL-8 and IL-10 (2000 and 400 pg mL^−1^, respectively) were detected vs. the untreated control (Figure 3c). In contrast, it was observed that PMA-THP-1 showed a different behavior during the 6 h to 24 h period (Figure 3c vs. Figure 2c) that corresponded to a decrease of the cytokine levels (5281 pg mL^−1^ vs. 55 pg mL^−1^), indicating that degradation was taking place.

### 2.4. Immunomodulation of *O*-2-Substituted (1-3)-β-D-Glucan Produced by P. parvulus 2.6 on Cytokine Production in the Co-Cultured Model of Gut Mucosa

Once the in vitro model of human intestinal mucosa was set up by the induction of inflammation of PMA-THP-1 with the LPS from *E. coli* 026:B6, the effect of branched (1-3)-β-D-glucans was carried out upon inflammation. Two co-treatment end points were analyzed: one after 6 h (Figure 4a,b) and the other after 24 h (Figure 4c,d).

The results obtained, after 6 h of co-treatment, showed that the increased production of TNF-α and IL-10 by Caco-2 cells, mediated by the LPS–PMA-THP-1 interaction, was statistically significant (*p* < 0.05) and counteracted by treatment with each of the three (1-3)-β-D-glucans (Figure 4a). Furthermore, in the presence of the three polysaccharides, the levels of IL-8 produced by enterocytes were significantly higher than those detected in the untreated control (*p* < 0.05). Moreover, only EPS L was able to significantly reduce the amount of IL-8 vs. LPS treatment with a *p* = 0.0015. With respect to PMA-THP-1 macrophages, only laminarin showed a higher degree of IL-8 reduction (*p* = 0.01) vs. LPS (Figure 4b), even though EPS L and EPS P achieved significant values with a *p* = 0.0077 and a *p* = 0.01, respectively. Furthermore, it was shown that co-treatment with LPS and either EPS L or EPS P potentiated the induction of IL-10 production by macrophages, with a *p* = 0.001 in both cases (Figure 4b), an effect also observed in the absence of LPS (results not shown).

Overall, the analysis of the in vitro model of gut intestinal mucosa demonstrated an anti-inflammatory effect of the co-treatment with (1-3)-β-D-glucans after 6 h, with a moderate decrease of inflammatory parameters (TNF-α and IL-8) and an increase of the anti-inflammatory marker (IL-10) in PMA-THP-1 cells, indicating a signal transduction of polysaccharide–Caco-2 interaction with macrophages. Moreover, these results indicate that the three (1-3)-β-D-glucans are able to diminish the inflammatory effect on Caco-2 cells by reducing the concentration of TNF-α, and therefore IL-10, while only EPS L reduced the chemotaxis mechanism orchestrated by IL-8. On the other hand, EPS L and EPS P but not laminarin diminished the concentration of TNF-α in PMA-THP-1, augmented the concentration of IL10, and reduced IL-8 together with laminarin, indicating a positive effect at the innate immune system level.

When cells were incubated for 24 h, Caco-2 co-treatment with any of the polysaccharides (Figure 4c) did not lead to a reduction of TNF-α determined by LPS–PMA-THP-1, as previously observed in cells treated during 6 h (Figure 4a). Furthermore, only EPS L and EPS P were able to inhibit IL-8 levels produced in response to the LPS-induced effect, showing a significant decrease with a *p* = 0.0078 and a *p* = 0.01, respectively (Figure 4d). However, no significant increase in IL-10 levels was observed for Caco-2 cells in co-treatments with LPS nor for each of the three (1-3)-β-D-glucans (Figure 4d).

Focusing the analysis on the PMA-THP-1 cells, co-treatment with the different polymers led to a slight decrease of the TNF-α levels (more than 6800 pg mL^−1^) caused by the stimulation with LPS (Figure 4d). Moreover, the levels of IL-8 synthesized by PMA-THP-1 cells were significantly elevated (approximately 2000 pg mL^−1^) in most of the samples (Figure 4d). Only EPS P resulted in a significative decrease vs. the control LPS (*p* = 0.03). Regarding IL-10, PMA-THP-1 cells treated with LPS secreted IL-10 levels (400 pg mL^−1^, Figure 4c) 10-fold higher than levels secreted by enterocytes (Figure 4d).

### 2.5. Effect of *O*-2-Substituted (1-3)-β-D-Glucan Produced by P. parvulus 2.6 on Gene Expression Profiles

In order to validate and complement some of the results obtained from the cytokine profiling secreted by the in vitro gut mucosa model, we evaluated the relative gene expressions of IL-10 and IL-8 in both Caco-2 (Figure 5a,c) and PMA-THP-1 (Figure 5b,d) cells at 6 h (Figure 5a,b) and 24 h (Figure 5c,d). After 6 h of treatment, analysis of IL-10 gene expression in Caco-2 cells (Figure 5a) revealed an inducing effect at the transcriptional level due to both EPSs, with respect to the untreated control (*p* < 0.05), but this effect was not observed after laminarin exposure. Likewise, all co-treatments stimulated via LPS–PMA-THP-1 interaction showed a significant difference with an increase of approximately 2-fold vs. the untreated control (Figure 5a). In addition, co-treatment with EPS P showed significantly increased IL-10 vs. LPS treatment with *p* < 0.05 (Figure 5a).

Indeed, treatment with either bacterial EPS (though not with laminarin) was able to significantly (*p* < 0.05) activate IL-8 gene expression (Figure 5a). A gene expression induction was also detected with the three (1-3)-β-D-glucans (45–60-fold) vs. the untreated control, and upon co-treatment with LPS, this stimulation was higher than that detected in treatments with either EPS (4–8-fold) and with stimulation with LPS alone (approximately 20-fold) (Figure 5a).

Regarding PMA-THP-1, a tendency was observed to reduce IL-10 expression levels in response to the treatments with the three (1-3)-β-D-glucans alone (Figure 5b), this being significant for EPS P and laminarin (*p* < 0.05). The stimulation with LPS triggered a high increase in the expression levels of this interleukin coding gene (Figure 5b), one that is statistically significant (*p* < 0.05) with respect to untreated samples. Moreover, in the case of treatment with EPS L and laminarin, these values were significantly higher than the control stimulated only with LPS (*p* < 0.05) (Figure 5b). IL-8 expression levels were found to be unaltered after treatments (Figure 5b). Only upon stimulation of macrophages with LPS was overexpression of the coding gene of this interleukin triggered (100–150-fold) vs. untreated control. An increasing trend was observed with co-treatment with the three (1-3)-β-D-glucans (Figure 5b). In line with the cytokine profiling results, LPS–PMA-THP-1 showed the ability to modulate gene expression transcription, indicating a signal transduction activation between macrophages and enterocytes.

After 24 h treatment, Caco-2 cells showed a slight reduction in IL-10 expression (Figure 5c) with either EPS, especially with EPS L (*p* < 0.05); levels were also significant with respect to the sample stimulated with LPS (*p* < 0.05) (Figure 5c). The (1-3)-β-D-glucan-Caco-2 and LPS–PMA-THP-1 co-treatment caused an increase in IL-10 expression levels by Caco-2 compared to the untreated samples (*p* < 0.05) and with higher levels than those detected in the sample stimulated with LPS (*p* < 0.05) (Figure 5c). These values were higher than those observed after 6 h of co-treatment (Figure 5a).

IL-8 expression was significantly reduced (2–3-fold) by treatment with the (1-3)-β-D-glucan. In addition, the elevated induction of IL-8 gene expression detected after 6 h treatment was not observed after 24 h. Only a significant increase (2-fold) with respect to the untreated control was observed with laminarin (*p* < 0.05).

Analysis of PMA-THP-1 cells showed that IL-10 expression values were not significantly affected by the treatments with any of the two EPSs; only in the case of laminarin treatment was a significant reduction (*p* < 0.05) observed in the levels of this transcript vs. the untreated control (Figure 5d). In addition, after 24 h of LPS stimulation or co-treatment (Figure 5d) the high induction (20- to 40-fold) detected at 6 h) was not observed (Figure 5b). However, only laminarin in co-treatment produced a significant decrease in the levels of the transcript encoding IL-10 (*p* < 0.05), compared to the untreated control. 

Treatments for 24 h with the (1-3)-β-D-glucans had no effect on IL-8 expression levels in PMA-THP-1 macrophages (Figure 5d). However, this situation was reversed upon activation with LPS, a situation in which their values significantly increased (approximately 10-fold) (*p* < 0.05), although the levels found in the co-treatments were similar to those detected in the LPS-treated cells, (Figure 5d).

## *3.* Discussion

Caco-2 and THP-1 (treated with PMA) differentiated to enterocytes and macrophages, are commonly used to model the conditions of the intestinal mucosa [35] and the associated human innate immune system in vitro [18]. Here, we have used this model to evaluate the immunomodulatory effects of two *O*-2-substituted (1-3)-β-D-glucans synthesized by the pediococcal GTF glycosyl transferase in *L. lactis* and of *P. parvulus*, in comparison with laminarin, another (1-3)-β-D-glucan substituted in the *O*-6 position and synthesized by the brown alga *Laminaria digitata.*

These morphological characteristics (confluence and formation of a monolayer with strong intercellular junctions) were assessed through optical microscopy, verifying the differentiation of the epithelium (results not shown), and through the TEER value. This study focused on determining the effects that the treatment with these polysaccharides could have on the modulation of the inflammatory response, mediated by PMA-THP-1, and triggered by the *E. coli* LPS in both cell types, enterocytes and macrophages. We also investigated how the response could be related to the type of polysaccharide and how it could affect the duration of the co-treatment. All this scientific evidence could be useful in order to determine the roles of bacterial β-glucan or laminarin in the prevention of chronic diseases or immune system activation to avoid stress immunodepression.

The bioactive candidates would be in contact with the intestinal microvilli to exert their function; therefore, we analyzed, by using the XTT assay, the possible cytotoxic effects of the (1-3)-β-D-glucan on the intestinal barrier, assessing the cellular metabolic rate, both in the monolayer undergoing differentiation (a situation similar to an intestinal ulcer, in which epithelial cells with a high proliferation rate migrate to repair tissue damage) (Figure 1a), as well as in a differentiated monolayer (simulating the normal conditions of the intestinal lumen, in which the cells lose their proliferative capacity) (Figure 1b). In both cases, the treatments with each of the three polysaccharides (at the concentrations studied) showed no cytotoxic effects and therefore did not alter the functional balance of the enterocytes. Likewise, no cytotoxic effect was observed on PMA-THP-1 cells treated with branched (1-3)-β-D-glucans (Figure 1c).

Once the cytotoxic effect of the three (1-3)-β-D-glucans was discarded, the in vitro model of inflamed gut mucosa was tested by stimulating PMA-THP-1 cells with the *E. coli* 026:B6 LPS in order to assess the mechanism of signal transduction determined by this LPS. Both PMA-THP-1 and Caco-2 cell lines positively responded to this stimulus by secreting high levels of the TNF-α pro-inflammatory cytokine (Figure 2 and Figure 3).

The Caco-2 cell line has been used in previous works to study intestinal inflammation induced by IL-1β or TNF-α cytokines, which stimulate the secretion of IL-8 (a marker of epithelial inflammation) [36,37]. In the current work, the LPS–PMA-THP-1 treatment resulted in an increased secretion of these proinflammatory cytokines (Figure 2b), which was able to stimulate enterocytes through the basal membrane [36]. In addition, Caco-2 cells differentiate autonomously to polarized enterocyte (with an apical portion, lumen, and a basolateral portion, lamina propria), and therefore, the response to stimulation may vary depending on the receptors expressed on both membranes [36,37]. In line with our results, it has been described that the addition of TNF-α to its basal membrane causes an increase in IL-8 secretion in both compartments [38]; therefore, the addition of LPS to the basolateral compartment seems to trigger the secretion of pro-inflammatory cytokines such as TNF-α and IL-8 in both compartments, the levels present in the lower compartment being the result of PMA-THP-1 and Caco-2 secretions (Figure 2b). This could also be a direct stimulation of the basolateral membrane of Caco-2 cells by the LPS. However, this was not the case, since addition of LPS to the lower compartment in the absence of macrophages, as well as direct stimulation of the apical membrane of the enterocytes with LPS in the presence of PMA-THP-1 cells, did not elicit an inflammatory response (data not shown). The membranes of the two cell lines carry TLR-4 receptors, which recognize LPS and trigger inflammation as well as the increase of gut barrier permeability in human models [19,39,40,41,42]. However, under our conditions, LPS alone was not able to activate the inflammatory response in Caco-2 cells monolayer. Therefore, our results indicate that LPS was necessary to trigger an inflammatory response in the in vitro intestinal mucosa model through the activation of PMA-THP-1 cells.

With the in vitro model established, we analyzed how the treatment of Caco-2 cells with (1-3)-β-D-glucans affected the modulation of the immune and inflammatory response in a context of intestinal mucosal inflammation, caused by the stimulation of macrophages by LPS. This condition resembles, for example, the inflammatory response that take place during the activation of immune system in chronic disease or pathogen infections. The overall effect detected with the two EPSs indicated a modulation of the response of the two cell lines, with a tendency to increase in both the concentration of the anti-inflammatory cytokine IL-10 and to reduce the levels of the pro-inflammatory cytokine’s TNF-α and IL-8 (Figure 4).

Moreover, the influence of exposure of Caco-2 cells to laminarin was analyzed because this polysaccharide is a *O*-6 substituted (1-3)-β-D-glucan, structurally related to the bacterial *O*-2-substituted (1-3)-β-D-glucan bacterial EPS. It has been described that the use of laminarin could be beneficial in the cure of inflammatory bowel diseases (IBDs), such as Crohn’s disease, and ulcerative colitis, in remission patients or those with a mild form of these diseases [43]. The anti-inflammatory effect of laminarin in the intestine of weaned pigs has been completely proven, as it was able to reduce the expression of pro-inflammatory cytokines such as IL-6 and IL1a and to increase IL-10 [44,45], as well as to increase mucin expression in the ileum and the colon. In line with our results, it should be highlighted the short-term effects (6 h) exerted by laminarin, modulating the in vitro model, mainly via IL-8, in both cell lines, determined by detection of an increase and decrease of this cytokine, respectively, in the Caco-2 cells and the PMA-THP-1 cells compartments. Regarding TNF-α and IL-10, laminarin shared the same effect observed for the EPS on Caco-2 cells (Figure 4). These differences could be the reflection of their different structures (EPS vs. laminarin). Another difference detected was the greater short-term effect of EPS L in reducing TNF-α production and increasing IL-10 levels in PMA-THP-1 cells (Figure 4, 6 h). While EPS P was successful in the long term (24 h), causing a reduction of IL-8 in Caco-2 cells (Figure 4), dissimilarities that could also be related to the different molecular masses of these EPS (9.6 × 106 Da for EPS P and 6.6 × 106 Da for EPS L) [46].

As macrophages are also responsible for the regulation of innate and adaptive inflammatory response, through the recruitment of different immune system populations such as neutrophils or T lymphocyte cells, it could be desirable to modulate their ability to secrete pro-inflammatory cytokines such as TNF-α or IL-8 in order to decrease the recruitments of these immune system populations and therefore reduce inflammation rate in the gut [47].

IL-8 modulation is important, as it is involved in the regulation of intestinal homeostasis by modulating luminal permeability (due to its effect of modifying the tight junctions along the intestinal epithelium) [37,48]. IL-8 is required to recruit the components of the immune system to the inflammation foci, and the EPS as well as the laminarin could play a role to avoid its production, which will damage the integrity of the intestinal tissue [48] or neutrophil recruitment [49]. Previously, Notararigo et al., 2021 [27] demonstrated in an ex vivo model of human biopsies that EPS L and EPS P decreased IL-8 secretion, as well as relative gene expression transcription. These data corroborated the previous results of Perez-Ramos et al., 2018 [28], where EPS P also diminished neutrophil’s activation in larvae zebra fish model, indicating the anti-inflammatory effect of this exopolysaccharide.

Incubation times during treatments appear to play and important role in promoting differential cell responses. The two end points selected here (6 h and 24 h) could represent different stages of the immune system response. In fact, 6 h co-treatment could represent the peak of the inflammation stage, while 24 h could represent the resolution of the inflammatory process. Macrophages showed an early response compared to epithelial cells (Figure 2 and Figure 4, 6 h vs. Figure 3 and Figure 4, 24 h), reflected in a reduction of inflammatory markers and an increase in anti-inflammatory markers. The (1-3)-β-D-glucans also generated a response in both types of cell without stimulation with LPS, which affected the production of cytokines (results not shown).

Gene expression levels were analyzed in enterocytes and macrophages by RT–PCR. The obtained results supported that treatment with the three (1-3)-β-D-glucans lead to a significant increase in the production and secretion of the anti-inflammatory cytokine IL-10 in both cell types (Figure 5), first in PMA-THP-1 and then, in a delayed manner, in Caco-2. By contrast, the analysis of IL-8 showed gene expression activation at 6 h, significant for Caco-2 (Figure 5a), followed by a normalization of expression at 24 h (Figure 5c). This activation could be related to the role of this cytokine in the inflammatory response of the intestinal epithelium, this interleukin being tissue-specific. In response to a stimulus at the basolateral membrane, Caco-2 cells should respond with an apical-basolateral secretion that would lead to a recruitment of immune cells to the lamina propria and enhanced luminal epithelial restitution [37], thereby favoring the mechanisms of inflammation resolution and tissue repair. Moreover, the expression of the cytokine thymic stromal lymphopoietin (TSLP), involved in the maturation of T lymphocytes, and specific to the epithelial tissue, did not show any variability at 6 h, while a slight decrease was observed at 24 h in Caco-2 cells (data not shown). These data could explain the anti-inflammatory behavior exerted by the three (1-3)-β-D-glucans on enterocytes, which in turn could be favorable to decrease the ongoing inflammatory process mediated by Caco-2 through LPS–PMA-THP-1 cells.

In summary, the immunomodulation of the (1-3)-β-D-glucans was confirmed in the in vitro model of intestinal gut mucosa after stimulation of PMA-THP-1 cells with *E. coli* LPS, which triggered the cellular response mechanisms in PMA-THP-1 and Caco-2 cells. Thus, LPS activates the molecular cascades of inflammation in both cell types, which, together with the polysaccharide treatment, would first induce an immediate response through activation of production and secretion of IL-8 by the epithelium. Nevertheless, LPS is also a TNF-α activator through TLR-4 activation, which can be observed in Figure 4b,d (6 h and 24 h exposures, respectively) in PMA-THP-1. The increase of IL10 in PMA-THP-1 (Figure 4d) could be a mechanism that takes place to reduce high rates of inflammation, but when PMA-THP-1 was treated with the three β-glucans, its values were lower, probably due to the ability of the glucans to reduce TNF-α production. These mechanisms of lowering inflammation would also activate the decrease of IL-8 in the epithelial cells after 24 h of treatment. Furthermore, the lack of activation of TSLP could be related to the anti-inflammatory nature of (1-3)-β-D-glucans, which could be of significance in treatment of mild forms of IBD, to ameliorate patient’s inflammation rate or in the development of functional food and/or as a food supplement to modulate immune system response.

## 4. Materials and Methods

### 4.1. Cell Cultures

Caco-2 cell line, obtained from the CIB-CSIC cell bank, was cultured in DMEM medium (Gibco, Waltham, MA, USA) supplemented with 10% FBS (Sigma-Aldrich, St. Louis, MO, USA), penicillin (Gibco, Waltham, MA, USA) at 100 U mL^−1^ and streptomycin (Gibco, Waltham, MA, USA) at 100 μg mL^−1^ and incubated at 37 °C in 5% CO_2_.

THP-1 cells were cultured in RPMI 1640 medium (Gibco) supplemented with 10% FBS (Sigma), penicillin (Gibco) at 100 U mL-1 and streptomycin (Gibco) at 100 μg mL^−1^ and incubated at 37 °C in an incubator at 5% CO_2_.

To obtain macrophage-like cells, THP-1 cells were treated for 72 h with RPMI 1640 medium supplemented with 40 nM PMA (Sigma–Aldrich, St. Louis, MO, USA). Cells were seeded into 24-well plates (Falcon, New York, NY, USA) at a concentration of 5 × 105 cells per well. After differentiation, unattached cells were removed by aspiration, and attached macrophages (PMA-THP-1) were washed twice with RPMI medium.

### 4.2. EPS P and EPS L Production and Purification

Bacterial *O*-2-substituted (1-3)-β-D-glucans synthesized by the GTF glycosyl transferase were used in this work. To these end, exponential cultures of *P. parvulus* 2.6, a LAB isolated from cider [50], and the recombinant strain L. lactis NZ9000[pGTF] [51], were used to produce EPS P and EPS L, respectively. *P. parvulus* 2.6 is a probiotic strain whose genome has been sequenced [30], and *L. lactis* NZ9000[pGTF] is a recombinant strain which expresses the *P. parvulus* 2.6 GTF glycosyltransferase. The EPSs were produced and purified, as previously described [52]. Briefly, after removal of bacterial cells by centrifugation, the EPSs present in the culture supernatants were precipitated with three volumes of ethanol. Afterward, the EPSs were further purified by dialysis and fractionation by size exclusion chromatography after resuspension in a 0.3 M NaOH solution. Then, the EPS alkaline solutions were dialyzed, as above. Finally, the EPSs were lyophilized and kept at room temperature until use. After the first and second lyophilization, the purity of the EPSs was tested fluorometrically using specific fluorescent staining kits for DNA, RNA and proteins, as previously reported [53]. Solutions of the purified polymers were prepared at 1 mg mL^−1^. No contaminants were detected, as previously described [27].

### 4.3. Cytotoxic Assay with XTT

To test the possible deleterious effects of (1-3)-β-D-glucan treatments on an in vitro transwell model with Caco-2 and PMA-THP-1 cells, viability was determined using the XTT (2,3-bis(2-methoxy-4-nitro-5-sulfophenyl)-tetrazolium-5-carboxanil) proliferation kit (Hoffmann-La Roche, Basilea, Switzerland), following the supplier’s protocol.

Cytotoxic conditions for Caco-2 cells were tested in two experimental models: cells with high proliferation rate (differentiating monolayer) and cells with no proliferative capacity (differentiated and polarized monolayer). Cells were seeded in 96-well plates with complete DMEM (Gibco) medium at a concentration of 9 × 104 cells in 100 μL per well, in the case of the differentiating monolayer, and at a concentration of 2 × 104 in 100 μL per well, in the case of the differentiated monolayer. In the first case, the cells were cultured for 48 h (37 °C, 5% CO_2_), while in the second case, incubation was prolonged to 10 days.

PMA-THP-1 macrophages were cultured for 24 h at a concentration of 1 × 105 cells per well in a 96-well plate in RPMI (Gibco)-supplemented medium.

For all conditions, cells were treated with (1-3)-β-D-glucan during 24 h at the concentrations range of 1–100 μg mL-1 diluted in Krebs medium (Sigma–Aldrich, St Louis, MO, USA); untreated cells were used as negative control. After 24 h treatment, 50 μL of a solution containing XTT 1 and XTT 2 reagents (in a 1:50 ratio) was added to each well to obtain formazan, and absorbance at 450 nm (A450 nm) was measured in a microplate reader (BioRad 680). The determination of cell viability was performed by quantifying the conversion of XTT to formazan after exposure of the samples to XTT (XTT1 + XTT2). This measurement is relative and is referred to the untreated control, which is assumed to be 100% cell viability.

### 4.4. Co-Cultured In Vitro Gut Model of Caco-2 and PMA-THP-1

To establish the in vitro model of intestinal mucosa, transwell inserts were used (Corning, New York, NY, USA) with polycarbonate membrane and with pores of 0.4 μm of diameter with a density of 1 × 106 cm^−2^ pores. These inserts were previously hydrated with 1.5 mL of medium in the upper part of the chamber (insert) and with 2.6 mL of medium in the lower part of the chamber and kept for at least 2 h in an incubator (37 °C, 5% CO_2_) before seeding the cells (according to instructions of the manufacturer). Caco-2 cells were seeded in the inserts with complete DMEM medium, to the concentration of 1 × 10^6^ cells mL^−1^. The medium was renewed every 48 h. To monitor the stage of cellular differentiation, the resistance of the monolayer was measured every two days with a voltmeter, Millicell ERS-2 (Millipore, Burlington, VT, USA), until it reached a value of approximately 350–390 ohm (Ω). To calculate the TEER (trans epithelial electric resistance value) of the monolayer, the obtained resistance was subtracted from the value of an insert without cells:TEER = (R sample − R Control) × Chamber Area = Ω × cm^2^

Three days before the TEER value reached its optimum, THP-1 cells were differentiated to macrophages with PMA on 6-well plate using a concentration of 1 × 106 mL^−1^ cells in supplemented RPMI medium, as described above. When both cell type were correctly differentiated, the insert with Caco-2 was positioned on top of the chamber with PMA-THP-1. The medium used for both cells lines was non-supplemented DMEM or DMEM supplemented with the tested (1-3)-β-D-glucan at the concentration of 100 μg mL-1 for Caco-2 and/or the lipopolysaccharide from *E. coli* 026:B6 (LPS) and 10 ng mL-1 for PMA-THP-1. The incubations were performed across 6 or 24 h (37 °C, 5% CO_2_). The three (1-3)-β-D-glucans evaluated in this model were EPS P, EPS L and laminarin from Laminaria digitata (Sigma–Aldrich, St. Louis, MO, USA).

### 4.5. Determination of Cytokine Levels

Cell culture supernatants were used to quantify levels of cytokines: pro-inflammatory (TNF-α and IL-8) and anti-inflammatory (IL-10). For the TNF-α and IL-10 the OptEIA (BD) kit was used, while the DuoSet kit was used for the IL-8 ELISA (RD Systems, Minneapolis, MN, USA), following the manufacturer’s specifications.

Briefly, for TNF-α and IL-10, a Maxisorp (Nunc) a 96-well plate was incubated throughout the night, independently with 100 μL of each antibody diluted according to commercial specification. The next day, the plates containing immobilized antibodies, were washed 3 times with 300 μL of washing solution phosphate-buffered saline (PBS), (137 mM NaCl, 2.7 mM KCl, 8.1 mM Na_2_HPO_4_, 1.5 mM KH_2_PO_4_, pH 7.2–7.4), 0.05% Tween-20 (Sigma-Aldrich, St. Louis, MO, USA)), then incubated during 1 h with 200 μL of blocking solution (PBS 1×, 10% FBS (Sigma-Aldrich, St. Louis, MO, USA)). The washing cycle was repeated, and then 100 μL standard and samples were incubated for 2 h. Plates were washed 5 times, and then 100 μL of detection solution, containing biotin and streptavidin, was added to the wells and incubated for 1 h. Plates were washed 7 times. Finally, 100 μL of 3,3′,5,5′-Tetramethylbenzidine (TMB) was added to each well, consisting of reagent A (hydrogen peroxide) and reagent B (TMB) in proportion 1:1. Plates were incubated for 20 min in obscurity. The reaction was stopped by adding 50 μL of 2 N H_2_SO_4_, and the absorbance (A450 nm) was measured in a microplate reader (Biorad 680). All reactions were performed at room temperature.

Regarding IL-8, the antibody was diluted in PBS. The 96-well plates were coated with 100 μL of the PBS buffer; the coverslips were sealed, and the plates were incubated overnight at room temperature to allow the antibody to adhere to the wells. The next day, the unbound antibody was aspirated, and each well was washed 3 times with 400 mL of washing buffer (0.05% Tween 20 in PBS, pH 7.2–7.4). After the last washing, the plate was dried by inversion onto a clean paper. Subsequently, the unspecific binding sites of the antibodies were blocked by adding 300 μL of blocking solution (1% BSA, Sigma–Aldrich, St Louis, MO, USA, dissolved in PBS, pH 7.2–7.4, filtered with a 0.2 μm membrane) and incubated at room temperature for at least 1 h. Then the plates were washed 3 times. The antigen–antibody binding was performed by adding to each well 100 μL of the sample (cell culture supernatant) or of the standard cytokine diluted in the dilution reagent (0.1 % BSA (Sigma–Aldrich), 0.05 % Tween 20 (Sigma–Aldrich) in 20 mM Tris buffer saline–Trizma base (Sigma–Aldrich), 150 mM NaCl (Sigma–Aldrich), filtered through a 0.2 μm membrane). Therefore, the solutions present in the wells were aspirated and the wells were washed 3 times with the washing solution. For the detection of IL-8, 100 μL of the detection antibody combined with biotin was added to each well and incubated for 2 h at room temperature. The remaining antibody was aspirated, and the 3 washing cycles were repeated. Then 100 μL of streptavidin was added to each well, and the plates were incubated for 20 min at room temperature in the dark.

Then the wells were washed 3 times. Finally, 100 μL of the substrate solution (1:1 mixture of reagent A and reagent B) was added to each well, and the plates were incubated for 20 min at room temperature and avoiding direct light.

The reaction was blocked by adding 50 μL of 2N H2SO4 to each well, and A450 nm was measured in a 680-plate reader (Bio-Rad, Hercules, CA, USA).

Cytokine concentrations were extrapolated using the regression line generated, from the absorbance values of a standard curve obtained in the same assay with known concentrations of commercial cytokines.

### 4.6. Total RNA Extraction and Analysis

Once the experimental treatments were completed, and after removing the supernatants, 600 μL of Trizol (Ambion, Waltham, MA, USA) were added to the Caco-2 cells, and 1 mL was added to the PMA-THP-1 cells. We kept the cells at 4 °C on ice, using RNase-free materials and environment and following the supplier’s instructions.

For total RNA extraction, cells were incubated for 5 min at room temperature and mechanically disaggregated through 1 mL micropipette tips. Then, 0.2 mL of chloroform (Sigma–Aldrich) was added to each mL of Trizol, vortexed for 15 s and incubated for 2–3 min at room temperature.

After incubation, samples were centrifuged for 15 min at 12,000× *g* and 4 °C in a 5415R centrifuge (Eppendorf). To precipitate the RNA, the aqueous phase was transferred to another microtube, and 500 μL of 100% isopropanol (Sigma–Aldrich, St. Louis, MO, USA) was added for each mL of Trizol used. The mixtures were kept at room temperature for 10 min and, after centrifugation for 10 min at 12,000× *g*, 4 °C, the supernatants were removed. The precipitates were resuspended with 1 mL of ethanol 75% for each mL of Trizol used. The samples were vortexed and centrifuged for 5 min at 7500× *g*, 4 °C. After removal of the supernatants, the sedimented RNAs were dried at room temperature for 5–10 min under vacuum. Total RNA preparations were resuspended in 25 μL of DNase/RNase-free ultrapure water and stored at −80 °C.

Once the total RNA was obtained, its concentration and integrity were evaluated in a Bioanalyzer 2100 (Agilent Technologies, Santa Clara, CA, USA) using the RNA 6000 Nano Chip kits. Only those samples in which the ribosomal RNAs 28S and 18S were in a ratio >1.5, and in which the RNA integrity number (RIN) was higher than 6 (according to the manufacturer’s instructions), were accepted for further usage.

### 4.7. Synthesis of cDNA from Total RNA Preparations

For the synthesis of cDNA from RNA, the kit “High-Capacity cDNA Reverse Transcription” (Applied Biosystems, Santa Clara, CA, USA) was used, applying the recommended protocol.

Microtube mixes contained: 25 μL of the 2 × RT master mix (containing the MultiScribe reverse transcriptase, the ribonucleotides and buffer required for cDNA synthesis) and 25 μL of an aqueous solution containing 2 μg of total RNA. cDNA was synthesized in a Mastercycler thermocycler (Eppendorf). At the end of the reaction, 50 μL of RNase-free water was added to each tube, which were then stored at −20 °C.

### 4.8. Analysis of Gene Expression by Quantitative PCR

Gene expression profiles were determined using quantitative real-time PCR (qPCR) with TaqMan probes (Applied Biosystems, Santa Clara, CA, USA), using the “TaqMan Fast Universal PCR Master Mix” (2X) kit (Applied Biosystems, Santa Clara, CA, USA) and the 7500 Fast Real time PCR System (Applied Biosystems). The expression of the genes encoding the interleukins IL-8, IL-10 and TSLP (thymus stromal lymphopoietin) was analyzed, using assay-on-demand probes, following the protocol recommended by the supplier. Triplicates of the assays were analyzed in a 96-well PCR plate; the reactions contained: TaqMan gene expression assay (for each cDNA sample), an endogenous or constitutive gene expression control gene (cyclophilin A, PPIA) and a negative control without cDNA (to assess possible nucleic acid contamination). The mixture was prepared for each gene: 10 μL of MasterMix, 1 μL of TaqMan probe and 4 μL of water. For each reaction, 15 μL of this mixture plus 5 μL of the cDNA (previously synthesized) were added to each well.

The qPCR results were analyzed with the Data Assist software (Applied Biosystems), considering the overall amplification plot of the genes on the plate, establishing the baseline and threshold values, and ending with the analysis of the results using the comparative threshold cycle (Ct) method: 2ΔCt. This method is based on the comparison of the Ct values of the treated samples with those of the untreated control (normalized against the end gene of constitutive expression).
∆∆Ct = ∆Ct treated − ∆Ct Control

### 4.9. Statistical Analyses

Data were statistically analyzed using GraphPad Prism version 8.0 (GraphPad Software, Inc., La Jolla, CA, USA).

Data underwent the Kolmogorov and Smirnov normality test. Significance for paired parametric values was obtained with a two-tailed t test for parametric data or with the Wilcoxon matched-pairs signed rank test for no parametric data.

Co-culture assays data were analyzed in two different ways:
Treatment vs. untreated control to assess the possible effects of the EPS and laminarin in a basal state vs. untreated control. Significant values were represented as *p**.Treatment vs. inflammation control: to assess possible inflammation-modulating effects of EPSs and laminarin vs. inflammatory state (LPS treatment). Significant values were represented as *p*✽.

## 5. Conclusions

The use of bacterial *O*-2-substituted-(1-3)-β-D-glucans as post-biotic food additives or as functional food components by in situ production during food fermentations could be effective to modulate the host immune system. Pre-clinical study, using an in vitro model of gut mucosa, demonstrated that its use as immunomodulator of innate immune system cells and epithelial cells could play a role in diminishing pro-inflammatory markers such as TNF-α and IL-8 and increasing anti-inflammatory mediators such as IL-10, not only during inflammation processes but indeed in steady state. Laminarin, another branched (1-3)-β-D-glucan, was shown to be effective in the resolution of the inflammation, especially at the early stage of this process, increasing IL-10 secretion and IL-8 reduction. Taken together, these results could be of interest to ameliorate immune system response in normal individuals, as well as in some chronic inflammatory states, when the relapses are under control by common treatments.

## Figures and Tables

**Figure 1 ijms-23-01527-f001:**
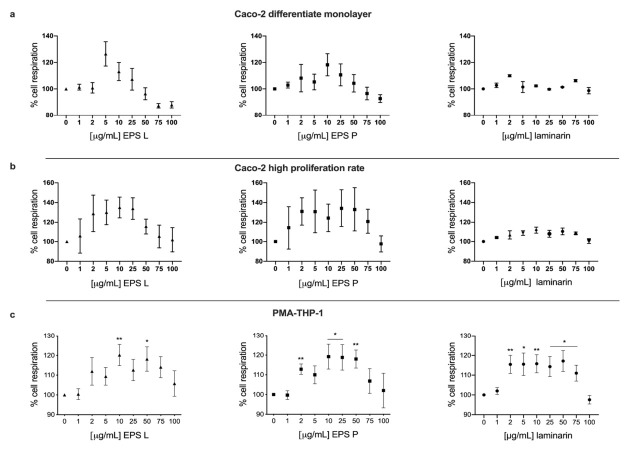
Influence of (1-3)-β-D-glucans on viability of Caco-2 cells and PMA-THP-1. The test was performed with the indicated polysaccharides and the XTT kit. Dot blot means and standard error of the mean (SEM) are represented. Data underwent the Kolmogorov–Smirnov normality test. Paired two-tailed t test was applied for parametric data, and the Wilcoxon test was performed for no parametric data. Caco-2 cells were seeded in a 96-well plate in two different conditions: (**a**) to a low concentration in order to test (1-3)-β-D-glucans in a growing monolayer with high proliferation rate, and (**b**) in a differentiate and polarized monolayer, when cells lost the ability to proliferate. (**c**) THP-1 cells were seeded in a 96-well plate and differentiated to macrophages with PMA during 72 h, and then treatments were tested. EPS L showed significant difference at the concentrations of 10 μg mL^−1^ (*p* = 0.0056), 50 μg mL^−1^ (*p* = 0.01); EPS P showed significant differences at the concentrations of 2 μg mL^−1^ (*p* = 0.0013); 10 and 25 μg mL^−1^ (*p* = 0.016 and *p* = 0.019, respectively); 50 μg mL^−1^ (*p* = 0.004); laminarin 2 μg mL^−1^ (*p* = 0.009); 5 μg mL^−1^ (*p*= 0.02); 10 μg mL^−1^ (*p*= 0.008); and 25, 50 and 75 μg mL^−1^ (*p* = 0.023, *p* = 0.11, *p* = 0.2, respectively). * *p* < 0.05; ** *p* < 0.01.

**Figure 2 ijms-23-01527-f002:**
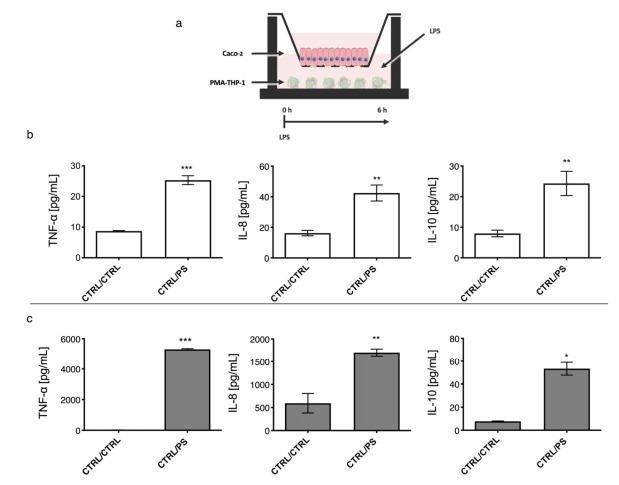
Stimulation of gut mucosa by LPS in a co-culture model during 6 h. Representation of co-culture model including Caco-2 cell in the upper chamber and PMA-THP-1 in the lower chamber (**a**). LPS treatment (addition of 100 μg mL^−1^) was administrated in the lower compartment across 6 h. Levels of TNF-α, IL-8 and IL-10 cytokines of the untreated (CTRL/CTRL) and treated (CTRL/LPS) samples in the upper (**b**) and lower (**c**) chambers are depicted. Data underwent a Kolmogorov–Smirnov normality test. Paired t test was performed for parametric data, and the Wilcoxon matched-pairs signed rank test was performed for no parametric data. Cytokine profiling demonstrated that after administration of LPS TNF-α, IL-8 and IL-10 showed statistically significative vs. untreated control. Caco-2 reached *p* = 0.0003 for TNF-α, *p* = 0.0020 for IL-10 and *p* = 0.0078 f for IL-8. PMA-THP-1 reached a *p* = 0.0001 for TNF-α, *p =* 0.01 for IL-10 and *p* = 0.0078 for IL-8. * *p* < 0.05; ** *p* < 0.01; *** *p* < 0.001.

**Figure 3 ijms-23-01527-f003:**
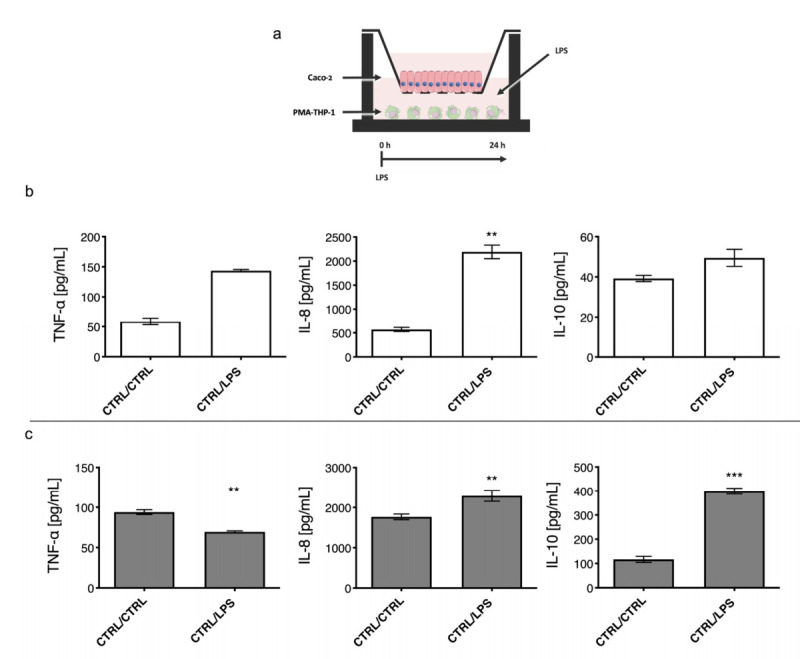
Stimulation of gut mucosa by LPS in a co-culture model during 24 h. Representation of the LPS-stimulated co-culture model (**a**). LPS treatment was administrated in the lower compartment during 24 h. Levels of TNF-α, IL-8 and IL-10 cytokines of the treated (CTRL/LPS) and untreated (CTRL/CTRL) samples in the upper (**b**) and lower (**c**) chambers are depicted. Data underwent a Kolmogorov–Smirnov normality test. Paired t test was performed for parametric data, and Wilcoxon matched-pairs signed rank test was conducted for no parametric data. Cytokine profiling showed that, after 24 h of treatment, only IL-8 reached values statistically significant for Caco-2 with a *p* = 0.0023. PMA-THP-1 cytokine production perpetuated throughout 24 h only for IL8 and IL-10 with a *p* = 0.0057 and a *p* = 0.0004, respectively, while TNF-α significantly reduced with a *p* = 0.001. ** *p* < 0.01; *** *p* < 0.001.

**Figure 4 ijms-23-01527-f004:**
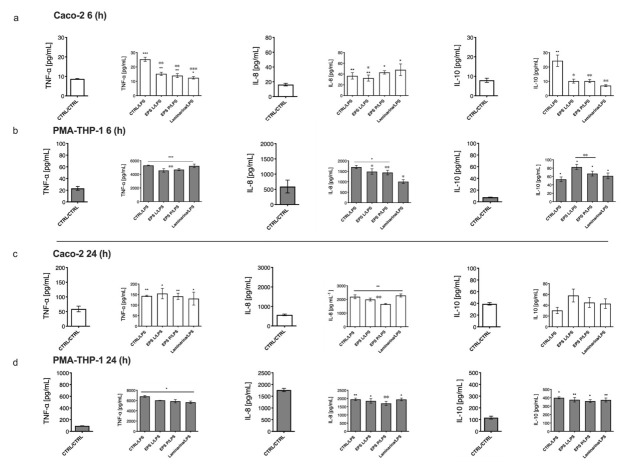
Effect of bacterial (1-3)-β-D-glucan and laminarin on co-culture model activated with LPS during 6 and 24 h. Barr plots are presented with mean and SEM. Data underwent a Kolmogorov–Smirnov normality test. For parametric data, a paired t test was performed; for no parametric data, the Wilcoxon matched-pairs signed rank test was performed. (**a**) Caco-2 upper compartment. TNF-α increased significantly after treatment with LPS, EPSs and laminarin vs. CTRL, with a *p* < 0.05. Indeed, TNF-α showed a significant decrease in comparison to LPS for the treatment of both EPSs and laminarin, with a *p* < 0.05. IL-8 increased for all treatments vs. CTRL, with a *p* < 0.05, while only EPS L decreased significantly vs. LPS with a *p* < 0.05. IL-10 increased significantly only for LPS vs. CTRL, with a *p* < 0.05, while the three (1-3)-β-D-glucans reduced its concentration vs. the LPS-treated control with a *p* < 0.05. (**b**) PMA-THP-1 lower compartment. TNF-α increased significantly in LPS and all (1-3)-β-D-glucans tested vs. CTRL and decreased after treatment with both EPSs vs. LPS with a *p* < 0.05. IL-8 increased significantly for LPS, EPS P and EPS L vs. CTRL, while all (1-3)-β-D-glucans reduced its concentration vs. LPS with a *p* < 0.05. IL-10 augmented significantly for all the treatments vs. CTRL as well as for EPS L and EPS P vs. LPS with a *p* < 0.05. (**c**) Caco-2 upper compartment. TNF-α achieved a statistical significance after LPS, EPSs and laminarin treatment vs. CTRL with a *p* < 0.05. IL-8 increased significantly after treatment administration vs. CTRL, with a *p* < 0.05, while it decreased for both EPSs vs. LPS with a *p* < 0.05. No significant effect was detected for IL-10, even though Especially EPS L showed a tendency to increase one-fold the CTRL and LPS values. (**d**) PMA-THP-1 lower compartment. TNF-α increased significantly after all the treatment administrations vs. CTRL, with a *p* < 0.05; the same was observed with IL-10. IL-8 showed a significant difference for LPS, EPS L and laminarin, with a *p* < 0.05, while only EPS P decreased significantly vs. LPS, with a *p* < 0.05. The overall data suggest that, after 24 h of stimulation, EPSs reduced LPS activation in both Caco-2 and PMA-THP-1 by the reduction of IL-8 concentration. Moreover, higher concentrations of IL-10 observed after 6h were able to counteract TNF-α levels in both cell lines.

**Figure 5 ijms-23-01527-f005:**
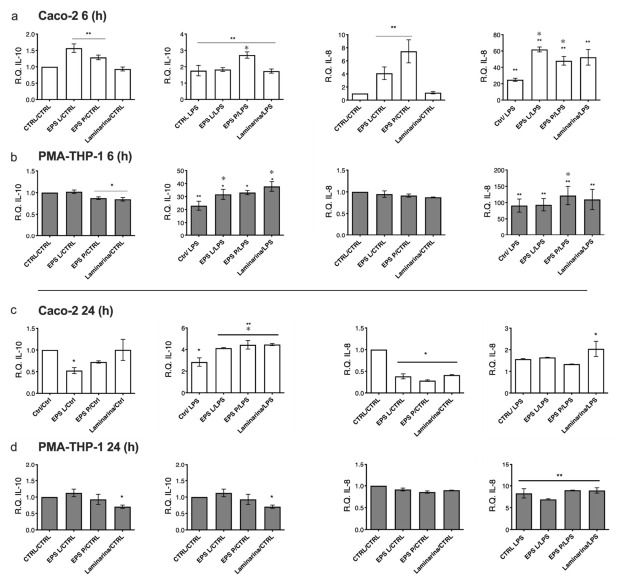
Relative gene expression quantification in the co-cultured model after 6 and 24 h of treatment. Data underwent a Kolmogorov–Smirnov normality test. For parametric data, a paired *t* test was performed; for no parametric data, the Wilcoxon matched-pairs signed rank test was performed. Co-culture assays data were analyzed in two different ways: Treatment vs. untreated control to assess the possible effects of the EPS and laminarin in a basal state vs. untreated control. Significant values were represented as *p**. Treatment vs. inflammation control: to assess possible inflammation-modulating effects of EPSs and laminarin vs. inflammatory state (LPS treatment). Significant values were represented as *p*✽. (**a**) Caco-2 gene expression profiling. Bar plot representation showed the effect on relative expression of (1-3)-β-D-glucans with and without LPS stimulation. EPS P, EPS L or LPS alone significantly increased IL-10, as well as the co-treatments (*p* = 0.02), vs. CTRL. Thus, only EPS P experimented a significant increase vs. LPS (*p* = 0.03). The same was observed for IL-8 transcriptional activation; both EPS and LPS alone, together with co-treatment, resulted in significantly altered results vs. CTRL (*p* = 0.002). Co-treatment with both EPS P and EPS L but no laminarin resulted in increased results vs. LPS (*p* = 0.002). (**b**) PMA-THP-1 gene expression profiling. IL-10 decreased significantly for EPS P and laminarin vs. CTRL (*p* = 0.03), but for LPS and co-treatment it increased significantly vs. CTRL (*p* = 0.0078). IL-8 increased after treatment with LPS and co-treatment vs. CTRL (*p* = 0.002), while only EPS P demonstrated an increase vs. LPS (*p* = 0.002). (**c**) IL-10 resulted in significantly reduced effects by EPS L treatment (*p* = 0.05) vs. untreated control; LPS together with the co-treatment resulted in augmented results, especially after (1-3)-β-D-glucan administration (*p* = 0.05 for LPS; p = 0.02 for all (1-3)-β-D-glucans) vs. untreated control. IL-8 was significantly repressed by the three β-glucans (*p* = 0.05). (**d**) IL-10 was reduced by laminarin (*p* = 0.05). Co-treatments also resulted in a significative increase (*p* = 0.05) vs. untreated control. IL-8 increased significantly only in co-treatment with LPS (*p* = 0.023); thus, EPS P showed a tendency to reduce its concentration.

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
