# Peer review of "Anti-Inflammatory Effect of an O-2-Substituted (1-3)-β-D-Glucan Produced by Pediococcus parvulus 2.6 in a Caco-2 PMA-THP-1 Co-Culture Model"

_ijms, 2022, doi:10.3390/ijms23031527_

Round 1

Reviewer 1 Report

Reviewer comments and suggestions

The structural similarity of exopolysaccharides (EPS) to active compounds such as laminarin, together with its ability to modulate the immune system prompted the investigator to study the comparison with them an immunomodulator of in vitro co-cultured Caco-2 and PMA-THP-1 cells.

The study results included XTT tests that revealed that all β-glucans were non-toxic for both cell lines and activated PMA-THP-1 cells metabolism. The immune modulation was similar by both polysaccharides. These features could be considered with the aim to produce function foods, supplemented with laminarin or with another novel β-glucan producing strain for inflammatory diseases

The paper was nicely written and needs some minor corrections before the final publication of the manuscript. Below are the comments for this paper to be incorporated in the revised version of the manuscript.

  1. Line 85, no need to use “on the other hand”
  2. For discussion: The first para needs to be short and discuss mostly a summary of 3-4 lines which were novel findings in this study. Try to reshuffle some parts, I think that would be enough to modify it
  3. Line 405-408 is there was any reason for this observation
  4. Line 413-414 please check if the sentence was correct, if yes try to explore it
  5. Line 456 Please explain the mechanism of the lowering of TNF alpha
  6. Almost all references need to be modified based on the MDPI journal format.

Author Response

  1. Line 85, no need to use “on the other hand”.

Answer. We removed “on the other hand” on line 85

  1. For discussion: The first para needs to be short and discuss mostly a summary of 3-4 lines which were novel findings in this study. Try to reshuffle some parts, I think that would be enough to modify it.

Answer: We have modified the first paragraph of the discussion, according to the suggestions of the reviewer (new Lines 319-325).

  1. Line 405-408 is there was any reason for this observation.

Answer: There was a mistake in the figure numbering. We meant the decrease that it was observed in figure 4 regarding EPS P. New lines 399-401.

  1. Line 413-414 please check if the sentence was correct, if yes try to explore it.

Answer: The sentences was corrected. New Lines 409-414

  1. Line 456 Please explain the mechanism of the lowering of TNF alpha

Answer: The sentence was modified. New Lines 456-462.

  1. Almost all references need to be modified based on the MDPI journal format.

Answer: The references have been corrected.

Reviewer 2 Report

This looks like an important manuscript in an area of human intestinal tract studies. In vitro transwell model that has been established by investigators can be a valuable milestone in future research. Authors demonstrated that use of bacterial O-2-substituded-(1-3)-β-D-glucans as immunomodulator of innate immune system cells and epithelial cells could play a role in diminishing pro-inflammatory markers.

I believe this manuscript deserves publication in International Journal of Molecular Sciences.

Author Response

We thank the reviewer for his kind comments.